# High-Density Lipoproteins Are Bug Scavengers

**DOI:** 10.3390/biom10040598

**Published:** 2020-04-12

**Authors:** Olivier Meilhac, Sébastien Tanaka, David Couret

**Affiliations:** 1Université de la Réunion, Inserm, UMR 1188 Diabète athérothrombose Thérapies Réunion Océan Indien (DéTROI), F-97490 Sainte-Clotilde, France; sebastientanaka@hotmail.fr (S.T.); david.couret@chu-reunion.fr (D.C.); 2CHU de La Réunion, Centre d’Investigations Clinique 1410, 97410 Saint-Pierre, France; 3AP-HP, Service d’Anesthésie-Réanimation, CHU Bichat-Claude Bernard, 75018 Paris, France; 4CHU de La Réunion, Neurocritical Care Unit, 97410 Saint-Pierre, France

**Keywords:** sepsis, periodontal disease, obesity, atherosclerosis, lipopolysaccharide, bacteria, virus, parasite, arthropods, HDL therapy

## Abstract

Lipoproteins were initially defined according to their composition (lipids and proteins) and classified according to their density (from very low- to high-density lipoproteins—HDLs). Whereas their capacity to transport hydrophobic lipids in a hydrophilic environment (plasma) is not questionable, their primitive function of cholesterol transporter could be challenged. All lipoproteins are reported to bind and potentially neutralize bacterial lipopolysaccharides (LPS); this is particularly true for HDL particles. In addition, HDL levels are drastically decreased under infectious conditions such as sepsis, suggesting a potential role in the clearance of bacterial material and, particularly, LPS. Moreover, "omics" technologies have unveiled significant changes in HDL composition in different inflammatory states, ranging from acute inflammation occurring during septic shock to low-grade inflammation associated with moderate endotoxemia such as periodontal disease or obesity. In this review, we will discuss HDL modifications associated with exposure to pathogens including bacteria, viruses and parasites, with a special focus on sepsis and the potential of HDL therapy in this context. Low-grade inflammation associated with atherosclerosis, periodontitis or metabolic syndrome may also highlight the protective role of HDLs in theses pathologies by other mechanisms than the reverse transport of cholesterol.

## 1. Introduction

High-density lipoproteins (HDLs) are regarded as cholesterol transporters in charge of reverse cholesterol transport from tissues back to the liver. They are associated with reduced risk of developing cardiovascular clinical complications, notably since the publication of the Framingham study, concluding that low HDL-cholesterol (HDL-C) levels were as much a risk factor for coronary artery disease (CAD) as high LDL-C [1]. However, HDL-C-raising therapies all failed at improving cardiovascular outcome, such as cholesteryl ester-transfer protein (CETP) inhibitors, which were very disappointing in phase III trials, for which the conclusion was “insufficient cardiovascular benefit for routine use" [2]. Supplementation therapies relying on an infusion of reconstituted HDLs did not significantly impact on atherosclerotic plaque regression [3] and still need to prove that they could be beneficial in reducing cardiovascular events [4]. In addition, recent Mendelian studies challenged the causal role of low HDL-C levels in cardiovascular diseases (CVD) [5]. In this context, HDL research in CVD resulted less attractive and is currently somehow abandoned. Old concepts have reemerged, coming from different medical specialties—in particular, from the infectious field. Rather than HDL-cholesterol concentration, HDL functionality is a parameter that should be evaluated as these lipoproteins display pleiotropic effects and may be modulated under pathological conditions. In this review, we will address the possible involvement of HDLs in infectious diseases, as well as their potential for therapeutic applications. 

## 2. HDLs and Innate Immunity: From Arthropods to Vertebrates

From an evolutionary point of view, lipoproteins are not only described as lipid transporters but also display important functions in many aspects of immunity. Invertebrate species have an open circulatory system called hemolymph, and their primary defense against microbial invasion may be related to circulating apolipoproteins in the first place and, then, by the activation of the coagulation system in response to a breach of their exoskeleton by either trauma or infection [6]. In *Galleria mellonella* (greater wax moth), the apolipoprotein III (also called apolipophorin III) has been reported to play an important role in immunity [7], acting as a pathogen recognition receptor stimulating the activity of defense peptides and shown to possess antimicrobial activity [8,9]. These properties have also been reported in the Chinese oak silkworm *Antheraea pernyi* [10]. In both vertebrates and arthropods, thrombus formation represents a basic mechanism of defense, allowing the trapping of whole bacteria or bacterial components such as LPS released from Gram-negative bacteria, flagellin, peptidoglycan or DNA unmethylated CpG motifs in order to limit their systemic dispersal [6,11]. Lipoproteins, and in particular, HDLs may play an important role in the case of a low-grade infection by enhancing the clearance of circulating bacterial material in order to limit thrombus formation to endothelial breaches (Figure 1). HDLs have been shown to display anticoagulant activity via enhancement of activated protein C/protein S [12]. Indeed, HDL levels are decreased in patients with venous thrombosis [13]. In addition to their direct action on the coagulation cascade, HDLs may also limit the prothrombotic effect of overactivated neutrophils. In response to LPS, neutrophils were shown to expel their DNA, forming a net that traps bactericidal proteins such as elastase, cathepsin G or histones aimed at limiting the spreading of bacteria [14]. These NETs (for neutrophil extracellular traps) also promote the formation of a thrombus [15] in collaboration with platelets via their Toll-like receptor 4 (TLR-4) [16] in order to confine bacteria. HDLs may then counterbalance the effect of neutrophils in case of bacteremia by quenching LPS and also by a direct action on neutrophil activation. Indeed, HDLs have been shown to limit neutrophil respiratory bursts [17] and subsequent activations—in particular, the release of myeloperoxidase and elastase in ischemic conditions [18] and in the context of atherosclerosis both in vitro and in vivo [19]. In this latter study, Murphy et al. reported that an infusion of reconstituted HDLs in patients with peripheral vascular disease presented a significantly attenuated neutrophil activation. Finally, HDLs may also modulate directly or indirectly platelet activation by limiting intraplatelet cholesterol overload or via their potential to increase NO and prostacyclin production by endothelial cells (potent inhibitors of platelet activation) [20].

## 3. HDLs: More Than Just ApoA1 and Phospholipid Nanoparticles

High-density lipoproteins are very complex macromolecular platforms organized following apolipoprotein A1 secretion by the liver and the intestine and the rapid recruitment of phospholipids, forming ApoA1-phospholipid nanoparticles that quickly evolve toward more mature particles enriched with cholesterol, cholesterol ester and triglycerides to form large-sized particles. HDLs are extensively remodeled in the blood stream via the action of lipid transfer proteins (LCAT, CETP and PLTP), as well as lipases (hepatic and endothelial lipases) but also due to lipid exchanges with cells via cholesterol transporters (ABCA1 and ABCG1) [26].

Whereas the role of HDLs in reverse cholesterol transport (RCT), from tissues back to the liver, is well-documented and widely accepted; these nanoparticles display pleiotropic effects [27] that may support their role in immunity and, in particular, in the host defense to micro-organism aggression. Following the development of "omics" at the beginning of the 21st century, including proteomics, lipidomics and the analysis of small RNA in HDL particles, more and more functions could be foreseen for HDLs. The notion of functionality has emerged rather than HDL-C concentration, which appears to reflect only one side of the story. Identification of new proteins and lipids in HDL particles supported new functions for HDL (i.e., antiproteases) and underlined the inappropriateness of actual HDL classifications. In fact, HDL particles are probably constantly remodeled in response to particular physiological or clinical situations, displaying different protein and lipid cargos. Thus, there may be HDL particles with distinct proteomic profiles and distinct biochemical functions.

### 3.1. HDL Proteome

In addition to ApoA1, HDL particles contain major apolipoproteins, including ApoA2, Apo-CI, Apo-CII, Apo-CIII and key enzymes involved in their remodeling. ApoA2 is the second-most abundant protein in HDL particles, but its role in septic conditions is not clear. Thompson et al. report that ApoA2 may increase the monocyte response to LPS, thereby playing a pro-inflammatory role [28]. Other authors demonstrated that both ApoA1 and AApoA2 limit neutrophil activation (decreased oxidative burst and interleukin-8 production) [22]. Apolipoprotein M is a 25-kDa protein contained in about 5% of HDL particles that can bind sphingosine 1-phosphate (S1P, see below), whose concentration is drastically decreased in sepsis and inflammatory conditions [29]. Apolipoprotein M (ApoM) was shown to limit LPS-induced acute lung injury [30], as well as mortality in LPS-treated mice [31].

Since the development of proteomic approaches and their blooming in early 2000, a large variety of proteins have been identified, being associated with HDL particles. Karlsson and colleagues identified 13 proteins by 2D-electrophoresis/mass spectrometry—of which, alpha-1 antitrypsin (AAT) and salivary alpha amylase had not been identified before [32]. Based on this publication, we confirmed that AAT was present in HDL fractions, either isolated by the classical method of ultracentrifugation or immunocaptured using an anti-ApoA1 column [33]. This acute phase protein is the natural inhibitor of elastase and could, in association with HDLs, prevent tissue damage caused by this potent proteolytic enzyme. HDL fractions could also be enriched with AAT and limited pulmonary aggression in a mouse model of emphysema [34]. Other proteins have been identified in HDL that could participate in their non-RCT, pleiotropic functions. Vaisar and colleagues have used a shogun proteomic approach to identify 48 proteins in HDLs isolated from healthy or coronary artery disease (CAD) patients by ultracentrifugation. Whereas 22 proteins were linked to lipid metabolism, 23 out of 48 belonged to the acute-phase-response protein family, known to be modulated under acute and chronic inflammation conditions (including AAT). Other proteins were related to the complement pathway, and its regulation, as well as specific ApoA1 complexes that help kill pathogens, suggested that HDLs may have a role in innate immune response.

More than 100 different proteins have been identified in HDLs, suggesting a multiplicity of functions for HDL particles [35]. However, all protein species cannot be present in one HDL particle or even in one subspecies, suggesting that only a fraction of the particles carry specific proteins [36]. 

### 3.2. HDL Lipidome

Lipids account for about half of the HDL particle mass and comprise more than 200 species identified, including phospholipids, cholesterol (free or esterified), sphingolipids and triglycerides [37,38]. Lipids contained in HDLs may display anti-infectious activity. For example, gangliosides have been used in reconstituted HDLs (rHDLs) to binds polymeric cholera toxin and protect cells from this biological toxin [39]. Other gangliosides presents in HDL particles may also be relevant to protection from infection but need to be further investigated [40]. Sphingosine-1-phosphate (S1P) is a bioactive lipid (product of sphingosine phosphorylation by sphingosine kinase or released from ceramide) mainly transported by HDLs in the bloodstream that regulates pathophysiological processes involved in sepsis progression [41]—in particular, proinflammatory cytokine release, endothelial permeability and vascular tone. For example, S1P activates eNOS (endothelial nitric oxide synthase) and inhibits MCP-1 (monocyte chemotactic protein-1) expression in endothelial cells [42,43]. Endothelial cells mainly express S1P receptor 1 (S1P1), followed by S1P2 and S1P3 [44]. Other S1P receptors, such as S1P5 receptors, may be important in the maintenance of the brain endothelial barrier function via stabilization of adherens junctions [45]. In pulmonary endothelial cells, S1P was shown to promote the endothelial barrier function by increasing the tight junction formation and cortical actin assembly, leading to a decreased permeability [46]. In vivo, intravenous or IP injection of S1P or of a pharmacological analog reduced pulmonary/renal vascular leakage and inflammation induced by intratracheal LPS administration [47]. In septic patients, the serum-S1P concentration was shown to be markedly decreased and is inversely associated with disease severity [41]. 

HDL lipidome is significantly modified in inflammatory conditions, including cardiovascular diseases—in particular, a decreased content of phospholipids and increased HDL triglycerides (for review, see [48]). Lipidomic studies have been performed in whole plasma of septic patients and in mouse models of sepsis [49]. For example, plasma lysophosphatidylcholine concentration was shown to be decreased on day 7 in nonsurvivors vs. survivors and was predictive of 28-day mortality in severe sepsis or septic shock patients [50]. To our knowledge, no specific HDL lipidome study has been undertaken in septic conditions.

### 3.3. HDL Small RNAs

In addition to proteins and lipids, HDL particles are able to carry nucleic acids such as small RNAs including miRNA, tRNA, snRNA, etc. [51]. MicroRNA (miR) are viewed as extracellular messengers that can be transported by lipoproteins and, particularly, by HDL particles [52]. HDL-associated miR profiles have been identified mainly in dyslipidemic and atherosclerotic patients [53]. Vickers’ group currently investigates small nucleic acids that could be transported by HDL particles under normal and pathological conditions. It is not excluded that HDLs may also transport nonhuman nucleic acid from viruses or bacteria.

## 4. HDLs Are LPS/LTA Scavengers

Most of lipoproteins have been reported to bind lipopolysaccharides, a major component of Gram-negative bacteria outer membranes. HDLs seem to be the more efficient for binding and inactivating different types of LPS [54]. 

Plasma phospholipid-transfer protein (PLTP) and cholesteryl ester-transfer protein (CETP) share similarities with LPS-binding protein (LBP) and with BPI (bactericidal/permeability-increasing protein) [55]. PLTP was first described as an enzyme involved in HDL remodeling by transferring phospholipids from VLDL/IDL and remnant chylomicrons to HDL particles. PLTP was then reported to transfer LPS to HDL in conjunction with LBP, leading to LPS neutralization [56]. Both LBP and PLTP can extract LPS from bacterial membranes and transfer it to HDLs [57]. HDL-associated LPS may enhance LBP-dependent LDL-HDL interactions, thereby forming a stable complex [58]. Whether these HDL-LDL complexes are more readily cleared from the circulation is not known. Data obtained from PLTP KO mice demonstrate that this enzyme is involved in LPS binding to HDLs and the subsequent LPS excretion into the bile, so-called "reverse LPS transport pathway" [55]. Whatever the mechanism, a wealth of experimental data suggests that HDLs bind and neutralize LPS. For example, Dai et al. clearly demonstrated that in vivo administration of fluorescently labeled LPS was found in the HDL fraction [59].

Gram-positive bacteria, albeit lacking LPS, are also important players in infectious diseases. Instead of LPS, their membranes contain lipoteichoic acids (LTA), which are amphiphilic molecules formed by a hydrophilic polyphosphate polymer linked to a neutral glycolipid, which is a major immunostimulatory component [60]. LTA are able to bind lipoproteins and, in particular, HDLs in a similar way as LPS do [54]. LTA can induce endothelial cell barrier dysfunction via TLR2 activation and the generation of reactive oxygen species [61]. LTA can also promote macrophage activation and the subsequent production of TNF-alpha. This is partially inhibited by HDLs in conjunction with LBP [62]. In vitro, ApoA1 was shown to bind and potentially neutralize LTA, thus reducing macrophage cytotoxicity in response to this immunostimulus. In vivo, ApoA1 administration in mice challenged by LTA significantly reduced inflammation in both serum and bronchoalveolar lavage fluid [63]. The mechanisms involved in LPS and LTA binding to HDL particles are not fully described. For LPS, the lipid A diglucosamine-phosphate region seems to be responsible for this association with HDLs [64]. LPS neutralization relies on LBP, which may form a complex between CD14 and LPS, favoring its binding to HDL particles and subsequent neutralization [65]. LTA is composed by a hydrophilic polyphosphate polymer linked to a neutral glycolipid, allowing its binding to HDL particles [66]. 

## 5. HDL and Sepsis

### 5.1. HDL-C Levels in Sepsis: What Do We Learn from Clinical Studies?

Increasing evidence suggests that both LDL-C and HDL-C concentrations are drastically decreased in critically ill patients [67] and, particularly, in septic conditions [68]. In this latter study, the authors showed an important decrease (up to 50%) of plasma cholesterol and a progressive recovery of subnormal levels of LDL-C and HDL-C after four weeks but, also, a replacement of serum amyloid A (SAA) by ApoA1, suggesting that HDL particles returned to their normal function. We have compared the lipid profiles between septic and trauma patients, two situations in which inflammation is exacerbated. HDL-C levels were markedly lower in septic patients relative to trauma patients, whose concentration was normal [69]. The mechanisms underlying this decrease in HDL plasma concentrations are not fully understood but could be due to an increased turnover paralleled by a reduced hepatic synthesis [70]. Endothelial leakage may also promote accumulation of HDLs in the extracellular compartment [71]. Whether HDL particles are involved in LPS/LTA clearance or whether they are only surrogate markers of multiple organ dysfunction needs to be investigated. A recent genetic study identified a rare variant of CETP (rs1800777-A), leading to decreased HDL-C concentration and an increased mortality in septic patients carrying this allele, associated with a more severe clinical picture [72]. HDL-C levels are globally inversely correlated with morbidity and sepsis severity. However, the link between HDL-C concentration and mortality is not a matter of consensus. Some studies clearly report that low HDL-C and ApoA1 concentration at admission are predictive of 18-30-day mortality and good markers of sepsis severity [73,74,75]. Others and we did not find a significant negative correlation between HDL-C and survival [68,76,77]. However, we report a poor outcome (SOFA score >6 or death) at day three for patients with lower HDL-C concentrations at admission [77]. Other studies have linked HDL levels to the increased risk of acute kidney injury [78] and disease severity in cirrhotic patients with severe sepsis [79].

### 5.2. HDL-Based Therapies in Endotoxemia and Sepsis Models

Based on the pleiotropic effects of HDLs, including anti-inflammatory, antioxidant and antiprotease properties, particularly beneficial for endothelial cells [43] but, also, due to HDL capacity to bind and neutralize LPS, many authors have tested their potential therapeutic effects in preclinical models and in clinical settings of endotoxemia. Most of the studies have used either reconstituted HDL particles or an ApoA1 mimetic peptide: D4F. 

D4F is an 18-aminoacid peptide with a similar structure to that of one of the eight amphipathic helical repeats of ApoA1, responsible for its lipid-associating property. This peptide is an improved class A amphipathic helix sharing structural similarities with ApoA1 but no homology with its amino acid sequence [80]. This particular peptide was optimized by adding four phenylalanine residues (4F) in order to improve its lipid-binding capacity. D4-F peptide mimics the major properties of HDLs, including LCAT activation and the inhibition of LDL-induced chemotactic activities [80,81]. It was also reported to improve glucose metabolism and to reduce hepatic inflammation in mice [82]. Reconstituted HDLs are made of ApoA1 isolated from human plasma and combined with soybean phosphatidylcholine. Initially produced by the Swiss Red Cross in Bern, they were subsequently a product of CSL Behring. Most studies conducted in mice consisted in the injection of LPS from *Escherichia coli* by either intraperitoneal (IP) or intravenous route (IV); administration of rHDL was performed a few minutes before endotoxemia. 

#### 5.2.1. Endotoxemia Models

The first study showing the beneficial effects of HDLs in sepsis is from Levine et al. in 1993, which nicely showed that transgenic mice with increased levels of HDLs were protected from lethal doses of LPS. They also injected reconstituted HDLs and a 18-mer peptide similar to D4F, showing that doubling the HDL concentration in mice resulted in a 3-4-fold increase in survival [83].

Different studies report the beneficial effects of 4F ApoA1 mimetic peptide administration in endotoxemia models, showing a reduction of inflammation associated with improved organ function. Kwon et al. report a decreased lung injury and improved survival in endotoxemic rats when the 4F peptide was administered 10 min after LPS IV injection [84]. Sharifov et al. also reported improved pulmonary function and decreased liver injury following 4F peptide treatment (by the IV route one hour after LPS IP injection). They also report that the 4F peptide inhibited neutrophil activation in response to LPS or serum from acute respiratory distress syndrome patients in vitro [85]. Two other studies using 4F peptides showed improvement in blood pressure and vascular reactivity, as well as improved cardiac performances in LPS-treated rats [59,86]. Casas et al. published two studies using rHDL in ex vivo and in in vivo models of endotoxemia in rabbits and showed that rHDL infusion (15 min pretreatment) thwarted the increase in TNF-alpha induced by *E. coli* LPS or whole-bacteria injections [87,88]. Finally, McDonald et al. and Zhang et al. published that rHDL with normal ApoA1 or ApoA1 Milano could attenuate multiple organ injury and dysfunction syndromes, probably via the inhibition of adhesion molecule expression and via anti-inflammatory and antioxidant effects [89,90].

#### 5.2.2. Cecal Ligation and Puncture (CLP) Model of Sepsis

CLP is a better model to mimic human sepsis relative to a simple LPS-induced endotoxemia and may be close to human sepsis progression, with similar hemodynamic and metabolic phases [91]. Cecum puncture mimics a polymicrobial sepsis model, leading to both Gram-positive and Gram-negative bacteremia. Using ApoA1 KO mice and ApoA1 transgenic mice, Guo et al. demonstrated that decreased HDL levels were associated with increased mortality in this CLP model, whereas ApoA1 overexpressing mice were more resistant to CLP-induced sepsis. HDL-mediated LPS clearance, modulation of corticosterone production and leukocyte recruitment were suggested to be responsible for resistance to sepsis in this model [92].

This model has been used to test both 4F peptide and reconstituted HDLs. A 4F peptide infusion decreased inflammation, protected kidney and heat injury and reduced CLP mortality in rats, potentially due to the prevention of sepsis-induced reduction in plasma HDLs [93,94]. Finally, we recently tested rHDL (CSL-111 from CSL Behring) IV injection two hours after sepsis in three preclinical models: CLP, intraperitoneal injection of *E. coli* (IAI76 strain) and *P. aeruginosa* pneumonia. Reconstituted HDLs were shown to limit mortality in CLP and *E.coli*-induced sepsis and provided a significant lung protection. HDLs were suggested to increase LPS clearance via the biliary route [95].

## 6. HDL and Viruses

Lipoprotein levels are modified during viral infections; for example, dengue severity is inversely correlated with total cholesterol and LDL-C but not to HDL-C levels, according to a recent meta-analysis [96]. Hepatitis C virus circulates in serum bound to triglyceride-rich lipoproteins, and HDLs may facilitate its entry into cells via SRB1 [97]. Similarly, it was demonstrated that ApoA1 could bind to the Dengue virus and increase its infectivity by promoting its entry into cells, also via SR-BI [98]. This HDL receptor may serve as a receptor for viruses (i.e., the hepatitis C virus) but also appears to be concentrated in cholesterol and sphingolipid-enriched plasma membrane microdomains (lipid rafts) [99], which may serve as a platform for virus entry [100]. 

In the human immunodeficiency virus (HIV) infection, a modification of HDL metabolism may occur, redirecting cholesterol to the ApoB-containing lipoproteins. This may alter the functionality of reverse cholesterol transport [70]. The relationship between virus infections and lipoproteins and especially HDLs is not as clear as it seems to be for bacteria. However, HDLs may display antiviral effects by neutralizing both DNA and RNA viruses, whether they have an envelope or not [101]. HDL-mediated antiviral activity could be due to ApoA1 interference with viral entry into the cell or during the fusion with the target cell [102]. HDLs may also induce direct viral inactivation [103]. Van Lenten and colleagues have reported that the D4F ApoA1 mimetic peptide could attenuate lung inflammation in mice infected with influenza A. This peptide also displayed antiviral activity, leading to a reduction of influenza titre by more than 50% PFU (plaque forming units/mg tissue) [104,105]. These studies suggest that ApoA1-based therapies may represent a potential therapeutic approach to fight influenza A-induced pneumonia. Furthermore, ApoA1 KO mice have increased inflammatory cell infiltration and impaired pulmonary vasodilatation [106], whereas ABCA1 KO mice (lacking this HDL-cholesterol acceptor) also display decreased pulmonary functions. HDLs and ApoA1 are acceptors for ABCA1 and ABCG1, respectively, which are notably expressed in various cell types of the alveolus, including alveolar epithelial type I and II cells (ATI and ATII), as well as alveolar macrophages [107]). HDL therapy in viruses with pulmonary tropism may thus be tested. 

## 7. HDL and Parasites

The main relationship between HDLs and their anti-parasite action relies on apolipoprotein L-I (ApoL1) [70]. In addition to their constitutive ApoA1, HDLs were reported to contain ApoL1 and haptoglobin-related protein (Hpr), which are important antimicrobial proteins providing protection from trypanosome infections [108]. These authors demonstrated that, individually, these proteins were 500-fold less toxic to Trypanosoma brucei than when they were assembled in HDL scaffolds. ApoL1 is present in the HDL3 fraction and was reported to be part of the trypanosome lytic factor-1 complex and responsible for its lytic activity [109]. ApoL1 can also limit infections with Leishmania and is able to kill this pathogen within phagolysosomes of the reticuloendothelial system [110]. 

## 8. Focus on HDL-Bound Paraoxonase 1

Paraoxonases (PONs) comprise a family of three enzymes that include PON1, PON2 and PON3, probably derived from a common precursor. PON1 is mainly transported by HDLs and has been reported to display antibacterial and antiviral activity [111]. PON activity is only screened by using synthetic substrates, whereas its native substrates remain largely unknown. The ability of PON1 to hydrolyze paraoxon is used to measure its activity, but this enzyme also possesses arylesterase and lactonase activity [112]. LPS injection in mice increased serum amyloid A (SAA) and decreased both PON1 and ApoA1 hepatic production [113]. In both preclinical models and in human sepsis, PON activity was shown to decrease, paralleled by a steep decline in HDL levels [114,115]. In other situations of bacterial infection, paraoxonase and arylesterase activity are markedly decreased: in pulmonary tuberculosis [116], during infection by *Helicobacter pilori* [117] or after exposure to *Chlamidia pneumoniae* [113]. PONs are able, via their lactonase activity, to hydrolyze and inactivate bacterial quorum-sensing molecules (QS). QS factors are secreted by Gram-negative bacteria in particular to regulate biofilm formation [118]. Paraoxonase activity is also decreased upon viral infection (for review, see [111]); this is reported for the hepatitis C virus [119], hepatitis B virus [120] or for the human immunodeficiency virus (HIV) [121]. PON1 was reported to participate in cholesterol efflux from the cell membrane to HDLs [122]; this may contribute to reduce the presence of cholesterol rafts that are necessary for viral infection. This phenomenon could influence HIV replication, which requires cellular plasma membrane cholesterol for its final assembly and entry into the cell. Finally, Raper et al. [123] demonstrated the presence of ApoA1, haptoglobin-related protein, PON1 and traces of ApoA2 in trypanosome lytic factors (TLF1) isolated from human serum. Whether PON1 plays a direct role in TLF1 activity needs to be investigated. Since most infection processes involve an oxidative stress, the antioxidant activity of PON1 associated with HDL particles may represent an important modulator of bacterial, virus or parasite development. 

## 9. HDL Remodeling during Inflammation

HDLs display important anti-inflammatory functions—in particular, by modulating the production of cytokine or leukocyte activation in response to different stimuli, particularly well-documented during the atherosclerotic process, but also under septic conditions [124]. For example, HDLs have been reported to limit inflammasome activation by cholesterol crystals by reducing NRF3 expression [125]. Whereas native HDLs may limit activation of the NLRP3 inflammasome pathway, leading to the production of IL-1b and IL-18, both oxidized LDL and oxidized HDL may have the opposite effect [126]. Anti-inflammatory actions of HDLs are addressed in the review by Murphy and colleagues [125]. We will focus on HDL remodeling during inflammation.

### 9.1. RCT Decreased in Inflammation

Chronic inflammatory states are thought to impair RCT and, thus, to contribute to atherosclerosis. Cholesterol efflux from macrophages, as well as HDL-C acceptor functions, are decreased under inflammatory conditions. Cholesterol elimination from the liver to the bile is also blunted via a decreased expression of biliary transporters ABCG5 and ABCG8 [127]. RCT function of HDLs in atherosclerosis will not be developed in this review.

### 9.2. Size and Composition

In both rodent models of endotoxemia and in humans, HDL particles undergo important structural modifications, including changes in size (get bigger) [69], have a reduced phospholipid content, potentially due to the activation of endothelial lipase [128] and sPLA2 [129,130] and show an increase in HDL-associated SAA. HDL particle size was evaluated by both NMR and native electrophoresis [131]. This latter study reports that, in addition to the depletion of pre-beta HDLs, a reduction of small/medium-sized particles was observed.

### 9.3. HDL in Low-Grade Inflammation: Periodontal Disease and Obesity

In addition to high plasma LDL-C leading to vascular wall lipid deposition, chronic inflammation has been reported to be the driving force of atherogenesis [132]. As stated in this review, “identifying the triggers for inflammation and unravelling the details of inflammatory pathways may eventually furnish new therapeutic targets”. The impact of periodontal disease (PD) on atherosclerosis is well-described from an epidemiological point of view, but evidence comes also from preclinical studies in rodent models, as well as observational and interventional clinical studies [133,134,135]. PD is a high-prevalence, noncommunicable disease (about 50% of the population is affected worldwide and 11.2% in its most severe form) [136]. HDL-C levels, HDL particle composition, structure, size and functionality have been extensively investigated in the context of atherosclerosis [35]. In low-grade, chronic inflammation such as in periodontal disease and in obesity, whole-bacteria and/or bacterial components are transclocated from the oral environment or from the intestine into the blood flow, potentially triggering inflammation in distant tissues such as within the arterial wall or adipose tissue. Intestinal permeability is increased in obese patients, leading to LPS and peptogycane translocation and, potentially, even whole bacteria could reach the adipose tissue [137,138]. In periodontal disease, 16S bacterial DNA has been found in atherosclerotic tissue such as in carotid or aneurysmal samples [133,134]. In patients with periodontal disease, paraoxonase activity and NO bioavailability were shown to be reduced, with increased oxidative stress; anti-inflammatory HDL function was blunted, leading to a deterioration of the HDL endothelial protective effect [139]. In a preclinical model of obesity, high-fat diet-fed mice (with saturated or monounsaturated fatty acids, respectively, SFA and MUFA) displayed an altered HDL proteome. Acute-phase proteins were differently associated on HDLs, reflecting an impaired liver-to-feces RCT in vivo in SFA-high-fat diet [140]. 

These studies suggest that low-grade inflammation potentially due to subclinical exposure to bacteria or bacterial components may significantly impact on HDL composition and function in periodontal disease and in obesity.

## 10. Conclusions

The different states of HDL particles are summarized in Figure 2, showing that under low-grade inflammatory conditions, such as diabetes, atherosclerosis or periodontal disease, both lipids and proteins may be modified, particularly by oxidation. During acute inflammation such as sepsis, HDL particles may become inflammatory (serum amyloid A may displace ApoA1). Finally, reconstituted HDLs composed by ApoA1 and phosphatidylcholines ("ApoA1 nanoparticles") may be used for therapy by intravenous injection. They could be enriched with hydrophobic, protective molecules.

In conclusion, HDLs have drawn the interest of the scientific community and pharmaceutical companies in the cardiovascular field for several decades but seem to be abandoned, since therapeutic approaches were shown to be disappointing. The pleiotropic effects of HDLs, including their antioxidant, anti-inflammatory, antiviral and endotoxin-scavenging properties, may ignite a new interest for theses lipoproteins in the field of infectious diseases. What if the primary role of HDL particles would be to scavenge and participate in the clearance of infectious material? What if the protective effects of HDLs in cardiovascular disease would rely on their anti-infectious property? 

## Figures and Tables

**Figure 1 biomolecules-10-00598-f001:**
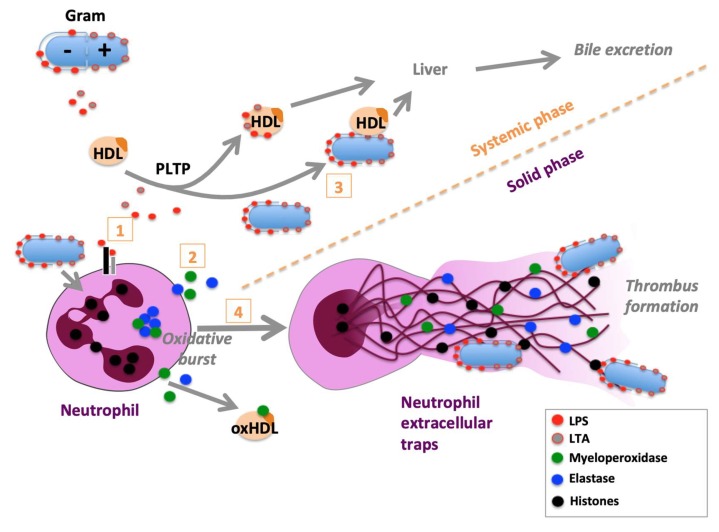
Role of high-density lipoproteins (HDLs) in the modulation of bacterial neutrophil activation. HDLs are able to bind and neutralize both lipoteichoic acid (LTA) [21] from Gram-positive bacteria and lipopolysaccharides (LPS) from Gram-negative bacteria via the action of the phospholipid transfer protein (PLTP) (1). These two bacterial components are potent activators of neutrophil activation, potentially leading to the formation of neutrophil extracellular traps (NETs). NETs consist in the extrusion of their nuclear content (DNA and histones), which forms a net associated with different proteins contained in neutrophil granules (such as myeloperoxidase and elastase). HDLs may also inhibit directly the oxidative burst (2) [22], leading to neutrophil activation and subsequent release of their granule content. A moderate neutrophil activation may be induced by LPS or bacterial phagocytosis without the formation of NETs. In this case, HDLs may be sufficient to promote the clearance of bacteria/bacterial material (3) via the liver and subsequent bile elimination (systemic phase). In these low-grade inflammatory conditions, ApoA1 may be oxidized by MPO and produce dysfunctional HDL particles (oxHDL) [23,24]. HDLs may also limit NET formation via the action of the soluble phospholipase A2 (4) [25]. In the case of a more sustained neutrophil activation, the production of NET is triggered, allowing the confinement of bacteria and promoting the formation of a thrombus (solid phase).

**Figure 2 biomolecules-10-00598-f002:**
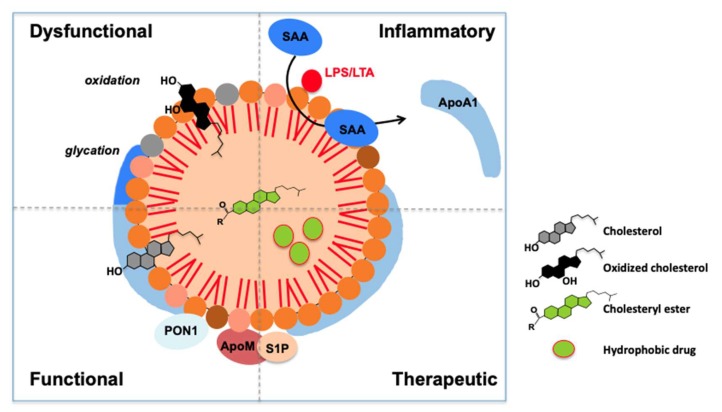
HDL particles in different states. Functional HDLs, containing different types of phospholipids, free cholesterol and esterified cholesterol, as well as different proteins such as ApoA1, ApoM associated with sphingosine 1-phosphate (S1P) and paraoxonase 1 (PON1). Dysfunctional particles are modified by different processes of oxidation or glycation on their protein and lipid moieties. Hydroxy- or ketocholesterol can be found, as well as myeloperoxidase-modified ApoA1 in low-grade inflammation observed in atherosclerosis or diabetes, for example. In inflammatory conditions, the serum amyloid A may replace ApoA1 and, in case of contact with bacterial material, lipopolysaccharide and lipoteichoic acid (LPS/LTA) may bind to HDL particles. Therapeutic ApoA1-nanoparticles may be used, taking advantage of the pleiotropic effects of ApoA1, in association with phospholipids (phosphatidylcholine is often used). These particles may be loaded with protective hydrophobic molecules for their delivery to inflamed tissues, in addition to the liver and kidney, naturally involved in HDL metabolism. SAA: serum amyloid A.

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
