# Peer review of "High-Density Lipoproteins Are Bug Scavengers"

_biomolecules, 2020, doi:10.3390/biom10040598_

Round 1

Reviewer 1 Report

This manuscript reviews the role of HDL as anti-inflammatory / anti-microbial particles, based on an evolutionary point of view and clinical evidences.

The text is easy to follow, most of the information is well supported by experimental evidences and the idea of HDL as anti-microbial particles is challenging and novel; it is time for new paradigms on HDL biological function, beyond the reverse cholesterol transport, and this manuscript contributes with fresh ideas in the field.

I have only some suggestions to improve the quality of the manuscript.

  1. Section 7. The discussion of this section focused on the capacity of paraoxonase on RCT; authors speculated that through the capacity of an increased cholesterol efflux, PONs are able to limit some viral and protozoal infections. In this context, what does mean “membrane metabolism”? How could the cholesterol efflux limit the availability of fatty acids for the amastigote shift of Trypanosome?  Why other apolipoproteins that promote cholesterol efflux (with higher efficiency than PONs) lack of such anti-viral/-protozoal effects? This section is highly speculative and should be supported by experimental reports. Otherwise, this section should be deleted in the corrected version.
  2. What about apo AII? Apo AII is the second most abundant apolipoprotein of HDL, evolutionally conserved, which role has been neglected. Could be this protein a modulator of the scavenge capacity of HDL? Could Authors include some information on this concept?

Reviewer 2 Report

General

The authors present a review covering many of the protective roles of high density lipoprotein (HDL) particles in inflammation or infectious diseases. The manuscript is well written and the relevant literature is sufficiently covered.

Specific comments

  1. Please refer to the role of HDL-associated apoA1 as MPO-binding protein (Fig. 1).
  2. For the interested reader it might be important to describe that HDL acts as a regulator of the NLRP3 inflammasome assembly process.
  3. Please describe the mode of LPS/LTA binding by HDL – protein vs. lipid domain – more detailed.
  4. Since the apoA1 mimetic D-4F is a promising pharmacological compound in preclinical models a more in-depth description would be warranted.
  5. HDL and virus binding: It might be worth noting that SR-BI can facilitate virus entry.
  6. In relation to SR-BI-mediated selective sphingolipid uptake it might be noteworthy that these SL-enriched plasma membrane microdomains serve as critical platforms facilitating virus entry.
  7. HDL lipidome: Please elucidate on potential roles of S1P/S1PR signaling pathways in EC barrier function, effects in different organs, means of pharmacological manipulation, and potential roles in ameliorating pneumonia/lung pathologies.

Minor:

Line 112 situations, line 145 nitric, line 210 studies, line 293 decreased, line 295 situations, line 300 decreased, line 328 reference format.
